# Immunogenicity and Safety of the Quadrivalent Adjuvant Subunit Influenza Vaccine in Seropositive and Seronegative Healthy People and Patients with Common Variable Immunodeficiency

**DOI:** 10.3390/vaccines8040640

**Published:** 2020-11-02

**Authors:** Mikhail P. Kostinov, Elena A. Latysheva, Aristitsa M. Kostinova, Nelly K. Akhmatova, Tatyana V. Latysheva, Anna E. Vlasenko, Yulia A. Dagil, Ekaterina A. Khromova, Valentina B. Polichshuk

**Affiliations:** 1Federal State Budgetary Scientific Institution, I.I. Mechnikov Research Institute of Vaccines and Sera, Malyi Kazenniy pereulok, 5a, 105064 Moscow, Russia; monolit.96@mail.ru (M.P.K.); anelly@mail.ru (N.K.A.); kate.khromova@mail.ru (E.A.K.); polischook@mail.ru (V.B.P.); 2Federal State Autonomous Educational Institution of Higher Education, I.M. Sechenov First Moscow State Medical University of the Ministry of Health of the Russian Federation (Sechenov University), Trubetskaya Str., 8/2, 119991 Moscow, Russia; 3National Research Center—Institute of Immunology Federal Medical-Biological Agency of Russia, Kashirskoe Shosse, 24, 115478 Moscow, Russia; ealat@mail.ru (E.A.L.); tvlat@mail.ru (T.V.L.); dogbreeder@bk.ru (Y.A.D.); 4Novokuznetsk State Institute for Advanced Training of Physicians—Branch Campus of the Russian Medical Academy of Continuous Professional Education, Prospect Stroiteley, 5, 654005 Novokuznetsk, Russia; VlasenkoAnna@inbox.ru

**Keywords:** adjuvanted QIV, immunogenicity, influenza, seropositive/seronegative, CVID, azoximer bromide

## Abstract

Background. Influenza prophylaxis with the use of quadrivalent vaccines (QIV) is increasingly being introduced into healthcare practice. Methods. In total, 32 healthy adults and 6 patients with common variable immunodeficiency (CVID) received adjuvant QIV during 2018–2019 influenza season. Depending on initial antibody titers, healthy volunteers were divided into seronegative (≤1:20) and seropositive (≥1:40). To evaluate immunogenicity hemagglutination inhibition assay was used. Results. All participants completed the study without developing serious post-vaccination reactions. Analysis of antibody titer 3 weeks after immunization in healthy participants showed that seroprotection, seroconversion levels, GMR and GMT for strains A/H1N1, A/H3N2 and B/Colorado, B/Phuket among initially seronegative and seropositive participants meet the criterion of CHMP effectiveness. CVID patients showed increase in post-vaccination antibody titer without reaching conditionally protective antibody levels. Conclusion. Adjuvant QIV promotes formation of specific immunity to vaccine strains, regardless of antibodies’ presence or absence before. In CVID patients search of new regimens should be continued.

## 1. Introduction

Although the novel coronavirus is currently causing a global pandemic, influenza continues to be one of the diseases capable of affecting the population, causing seasonal epidemics and sporadic pandemics, due to the emergence and spread of new influenza viruses. Vaccination has been and remains the most effective way to prevent it or, in some cases, leads to a mild, uncomplicated course of influenza. There is no doubt that trivalent influenza vaccines, consisting of two type A influenza strains (A/H1N1 and A/H3N2) and one influenza B lineage, have an epidemiological effect against these types of influenza viruses. It should be noted that trivalent vaccines contain only one of two antigenically distinct lineages type B influenza virus—Yamagata (YAM) or Victoria (VIC)—which are able to form a defense only from a specific vaccine strain. However, the results of long-term observations show that in 30–50% of seasons, the B virus lineage included in vaccine composition may not coincide with the actual circulating lineage among population, or both B lineages (YAM and VIC) can co-circulate in a different ratio [1,2].

Studies show that seasonal vaccines containing one of the B influenza virus lineages provide limited cross-reactive immunity against circulating alternative B lineage and only in 5 out of 10 seasons from 2001 to 2011 the vaccine and circulating virus coincide [3]. Despite the success of WHO think tanks, predicting which influenza B lineage will prevail over the coming season is problematic, and cross-protection by vaccination against non-vaccine line B is unclear [4,5]. Therefore, the epidemiological effect of vaccination of population against seasonal influenza can be increased by including strains of both influenza B lineages. Numerous studies have proved the safety and immunogenicity of quadrivalent inactivated influenza vaccines (QIV) compared to trivalent inactivated influenza vaccines (TIV) and antibody production in protective titers against additional B lineage, that indicates the advantage of vaccines containing two lineages of type B influenza virus due to the increase in the immune layer of population to influenza infection [6,7,8,9,10,11].

Of no less interest is the action of adjuvant QIV (aQIV) compared with adjuvant TIV (aTIV), in which MF59 is more often used as an adjuvant [12]. All these vaccines contain 15 μg of hemagglutiin (HA) of each influenza strain, a total of 60 μg of HA in aQIV and 45 μg of HA in aTIV. It was found that aTIV4 as well as aTIV demonstrated good clinical tolerance and formed similar immune response to common vaccine antigens in 65–74 years, 75–84 years and ≥85 years age groups, as well as in risk groups with concomitant pathology. As expected, aQIV excelled in the production of antibodies to both B lineages, that was not observed after aTIV use [9]. Similar results were obtained in other studies where aQIV was used [13,14].

One of the priority groups for the use of adjuvant influenza vaccines are people with various disorders in the immune system due to concomitant pathology or to the aging of the immune system. The other cohort of people requiring the use of such vaccines is those with innate defects of the immune system. A relatively favorable prognosis for the disease on the assumption of proper treatment and prophylactic measures have patients with defects in the humoral immunity. The growth rate of post-vaccination antibody titers is used as a diagnostic criterion in individuals with a common variable immune deficiency (CVID)—one of the most frequently diagnosed primary immunodeficiencies (PID).

Despite the fact that vaccination does not lead to the development of unusual adverse events and the disease impairment, post-vaccination immunity differs from the formation of the immune response of immunologically naive people. The search for methods of post-vaccination immunity improvement in CVID patients with changes of vaccination schedule (administration of at least two doses of vaccine or a booster dose administration after a certain time interval), as well as the development of vaccines with a high content of influenza antigens or adjuvant vaccines, are necessary to elicit an adequate seroprotective immune response. However, besides the well-known MF59^®^ adjuvant included in composition of adjuvant vaccines, in Russia, the family of influenza adjuvant vaccines with addition of azoximer bromide as an adjuvant has been broadly used for more than 20 years, which has proved its safety and immunogenecity in association with the vaccine influenza virus strains. The studies conducted on different age groups of healthy people and patients with different pathologies showed clinical and epidemiological effectiveness of these vaccines, and the azoximer bromide adjuvant significantly enhances the immune response to vaccination, that allows us to reduce the antigen content by three times compared to standard (5 μg). However, no studies of the effectiveness of influenza vaccines with the azoximer bromide used as an adjuvant in patients with CVID have been carried out.

The aim of the study was to conduct a comparative assessment of the immunogenicity of the licensed adjuvant quadrivalent inactivated subunit influenza vaccine in healthy individuals and patients with CVID.

## 2. Materials and Methods

### 2.1. Clinical Trial Design

The main goal of this study was to estimate the formation of specific antibodies for each strain of the influenza in the vaccine on the 21–22nd day after vaccination in both groups of healthy volunteers and CVID patients and to characterize the clinical tolerance of vaccination—to evaluate the development of local and systemic reactions (from the 1st to 8th day). The secondary objective was to assess the immunogenicity (day 85–90th) for strains A/H1N1, A/H3N2 and both B lineages (Yamagata and Victoria) in accordance with the CHMP criteria.

Stage IV (post-marketing), non-randomized, controlled, comparative study was conducted in two medical centers of Moscow (Russia)—Institute of Immunology of the FMBA of Russia and Mechnikov Research Institute of Vaccines and Sera—from October 2018 to March 2019. The protocol was approved by the local Ethics Committee of the Institute of Immunology. The study was conducted according to the Russian Federation National Standard Protocol ГOCTP 52379-2005 «Good Clinical Practice» and International GCP standards. The study was based on the ethical principles and recommendations of the WHO and the Russian Ministry of Health. All patients signed the informed consent for participation before the start the research.

### 2.2. Participants

Healthy adults aged 18–52 years were registered at a ratio of 5 (healthy): 1 (CVID) to receive aQIV, licensed in the Northern Hemisphere for the 2018–2019 influenza season, which contained two type A influenza strains and two lineages of type B influenza virus.

Volunteers were men (*n* = 20) and (*n* = 12) women aged 18–52 years who were healthy and did not have concomitant diseases (*n* = 32), determined by the medical history, medical examination and clinical opinion of the researcher.

Patients with CVID (*n* = 6) were men (*n* = 5) and woman (*n* = 1) aged 18–45 years, who were registered in the Department of Immunopathology of the Institute of Immunology, where they received regular replacement immunotherapy with intravenous immunoglobulin (IVIG), which are the basic therapy for patients with CVID.

The treatment of CVID is monthly life-long replacement IVIG therapy. All the important IgG antibodies presented in normal population are extracted from a large pool of human plasma from more than 1000 donors. Despite a large number of donors, IVIG does not contain antibodies against actual influenza virus in sufficient titer.

For the study period, the main requirement for patients with CVID was the absence of IVIG therapy for a total of 7 weeks. Instead of the scheduled IVIG administration, which is performed once every 4 weeks in average, one dose of the vaccine Grippol quadrivalent was administered to each participant, following 3 weeks (21–22 days) without basic IVIG therapy in order to determine more accurate values of the ability to synthesize their own antibodies. At 21–22 days after vaccination, patients with CVID were scheduled to undergo substitutional immunotherapy with IVIG in a standard dose of 0.4 g/kg. The next sampling of whole blood was 3 months-3 months and 1 week after vaccination on the background of regular replacement therapy. The entire post-vaccination period (90 days) CVID patients were under the supervision of the immunologist-researcher.

Inclusion criteria for healthy volunteers:Healthy people aged from 18 to 52 years without chronic bronchopulmonary, cardiovascular, rheumatological diseases, hepatic or renal impairment, metabolic disorders confirmed by anamnestic data or objective clinical examinationSigned informed consent.

Inclusion criteria for CVID patients:Confirmed diagnosis CVID in accordance with diagnostic criteria established by the European Society for Immunodeficiency Diseases (http://esid.org/WorkingParties/Registry/Diagnosis-criteria) and the American Academy of Allergy, Asthma and Immunology for the diagnosis and treatment of PID.Replacement immunotherapy with IVIG drugs no later than 28 days before vaccination and no earlier than 21 days after it, that is, a break between two subsequent administrations of immunoglobulins for at least 7 weeksSigned informed consent.

Non-inclusion criteria for healthy volunteers and CVID patients:A history of allergy to egg whites or any component of the studied vaccine.Symptoms of influenza or flu-like illness in the past 6 months.Vaccination against influenza within the last 12 months.Symptoms of acute infection or exacerbation of chronic disease at the time of vaccination or during 1 month before current vaccination.Glucocorticosteroid or other immunosuppressive therapy admission at the time of the study and 3 months before the start.Any medical interventions that have been identified as affecting the effectiveness of vaccination and receiving any type of vaccine within 1.5–2 months prior to inclusion in the study.Symptoms of enteropathy with protein loss in patients with CVID at the time of the study.Cognitive or behavioral or psychiatric disorders or alcohol abuse.

Exclusion criteria: violation of the conditions of the study protocol.

### 2.3. Vaccines

All participants received 1 dose (0.5 mL) of the vaccine, injection was made into the deltoid muscle on the first day of the study. The vaccine Grippol^®^ Quadrivalent (NPO Petrovax Pharm LLC, Russia) was registered with the Ministry of Health of the Russian Federation—RU No. LP-004951 from 07.23.2018. It was available for vaccination of people from 18 to 60 years old during the flu season 2018–2019. Grippol^®^ Quadrivalent contained four viral strains as recommended by the WHO for the 2018–2019 northern hemisphere influenza season for tetravalent vaccines: A/Michigan/45/2015 (H1N1)pdm09-like virus; A/Singapore/INFIMH-16-0019/2016 (H3N2)-like virus; B/Colorado/06/2017-like virus (B/Victoria/2/87 lineage); B/Phuket/3073/2013-like virus (B/Yamagata/16/88 lineage), isolated from the virus-containing allantoic fluid of chicken embryos and associated with the immunoadjuvant Polyoxidonium^®^ (international nonproprietary name—azoximer bromide).

The vaccine contained 5 μg of HA of each influenza strain, in total 20 μg, and azoximer bromide—500 μg (without preservative). Pharmacological action: anti-influenza, immunomodulatory. Pharmacological (immunobiological) properties: the vaccine causes the formation of a high level specific immunity against influenza. The protective effect after vaccination occurs after 8–12 days and lasts up to 12 months. Inclusion of the azoximer bromide immunoadjuvant with a wide range of immunopharmacological effects in the vaccine provides an increase in the immunogenicity and stability of antigens, immunological memory, and allows to reduce the amount of antigens in one vaccination dose without loss of its ability to form specific protective immunity [15]. Polyoxidonium as an adjuvant is well-established in different in vivo and in vitro studies, especially among patients with conditions followed by immune suppression [16,17,18]. Different studies demonstrated good tolerability of adjuvanted vaccine and no effect on fetus ant children development [19,20,21,22,23,24].

### 2.4. Immunogenicity Assessment

Immunogenicity was assessed by hemagglutination inhibition (HI) analysis performed on serum samples collected before and 21–22 days after vaccination by measuring antibody titers against influenza strains included in the vaccine. To remove nonspecific inhibitors of hemagglutination, test sera were incubated at 37 °C overnight (19 ± 1 h) at a 1:4 dilution with receptor-destroying enzyme (RDE; Denka Seiken, Tokyo, Japan) followed by a 30-min inactivation step at 56 °C and further dilution to 1:10 with phosphate-buffered saline. HI assay was performed with 0.5% chicken RBC and 4 hemagglutination units of antigens. Antigens for HI assay were provided by Smorodintsev Research Institute of Influenza (WHO National Influenza Centre of Russia, Saint-Petersburg). The primary analysis of the assessment of the immunogenicity of aQIV in healthy and patients with CVID was performed 3 weeks after vaccine administration, and the secondary analysis after 3 months (85–90 days), only in the patient group in accordance with the protocol. The 3 months control point is explained by the possibly delayed vaccine immunogenicity in CVID patients.

Depending on the initial antibody titers, healthy volunteers were divided into two groups—seronegative (≤1:20) and seropositive (≥1:40) to different influenza strains included in the vaccine. Vaccine immunogenicity was evaluated according to the last revision of the Guideline on clinical evaluation of vaccines from 26 April 2018 of the Committee on Human Medicinal Products (CHMP) criteria for adult patients: the percentage of vaccinated people with either a pre-vaccination HI titer <10 and a post-vaccination HI titer ≥40 or a pre-vaccination HI titer ≥10 and a ≥4-fold increase in HI titer on the 21 day after vaccination should be >40% (seroconversion criterion); increase in the mean geometric titer of HI antibodies on the 21 day after vaccination compared to baseline should be >2.5 (geometric mean rate (GMR) criterion); and the percentage of vaccinated people with HI titer ≥1:40 on the 21 day after vaccination should be >70% (seroprotection criterion).

The effectiveness of influenza vaccination with QIV in patients with CVID due to the small amount of participants was evaluated individually.

### 2.5. Safety Assessment

All participants were observed for 30–45 min after each immunization to monitor immediate adverse reactions. They were provided with study diaries, as well as constant communication through a daily telephone survey recording the frequency of local and systemic reactions within 7 days after vaccination, concerning CVID patients—14 days more, recording well-being. During that period of time, 4/6 had some problems with the provision of monthly immunoglobulins (2–3 weeks of drug delay). So it can even be said that CVID patients have benefited during the period of insecurity without IVIG therapy. Moreover, before the study, it was discussed that in the event of an infectious disease within 3 weeks of supervision after vaccination, we were ready to immediately administer a basic individual dose of immunoglobulins for each of these 6 participants. Fortunately, despite a long period without conducting basic IVIG therapy, none of the patients with CVID did not require its administration before the control points due to the absence of any symptoms of acute respiratory infections.

Local reactions were assessed by the incidence of pain at the injection site, erythema, compaction, and edema. Systemic reactions included chills, malaise, myalgia, arthralgia, headache, nausea, sweating, coughing and fever (axillary temperature ≥38.0 °C), as well as the use of analgesics or antipyretic drugs. The diameter of local reactions was classified: from 1 to 10 mm (no reaction), from 11 to 25 mm (mild), from 26 to 50 mm (moderate), from 51 to 100 mm or >100 mm (severe). Clinical observation was carried out for vaccinated both healthy and patients with CVID with registration of unusual events from the 1st to 21–22nd days.

### 2.6. Statistics

For the intergroup comparison of qualitative characteristics (levels of seroprotection and seroconversion), the Chi-Square test was used; in cases of cells in the table with expected frequencies of less than 5%, the exact Fisher test was used. Comparison of qualitative characteristics in related samples (in the dynamics between control points) was carried out using the McNemar criterion. Descriptive statistics of qualitative characteristics are represented by the fraction, 95% confidence interval of the fraction calculated by the Clopper–Pearson method, the absolute number of subjects with the studied characteristics in total number of groups (n/N).

Descriptive statistics of quantitative characteristics are represented by the geometric mean and its 95% confidence interval. To apply the statistical criteria the initial quantitative data were pre-logarithmized and checked for compliance with the normal distribution (the Shapiro–Wilk test was used). The check showed that all the pre-logarithmized data correspond to the normal distribution. To compare two independent groups by quantitative criteria, the Student criterion was used (in the absence of equality of variances, which was checked by the Livin test, the Student criterion with the Welch modification was used). Comparison of quantitative characteristics in related groups (in the dynamics between control points) was carried out by the Student criterion for related samples. Calculation of criteria for quantitative characteristics was carried out on logarithmized data. The analysis assumed a comparison between the values of characteristics at the control point of 1 month and the initial level and control points of 3 months and 1 month; if a statistically significant difference for 1–3 months was detected, the values of characteristics at the control point of 3 months was compared with the initial level. All calculations were carried out in a freely distributed statistical environment R (v.3.6), the “stats” package (v.3.6.2) was used.

## 3. Results

### 3.1. Safety of Vaccination

All participants completed the study without developing serious post-vaccination reactions. Among 32 healthy volunteers, local reactions after vaccination were registered in two of them and were characterized by local mild hyperemia which persisted for 2 days. Systemic reaction—malaise—was observed in one person 3 h after vaccination and persisted throughout the day. Observed post-vaccination events did not require prescription of any medicine.

In patients with CVID, local reactions in the form of hyperemia (50 mm in diameter) and edema appeared 2 h after the injection and persisted for 3 days after vaccination. Only one systemic reaction (chills) was registered at the day of vaccination without subsequent temperature increase, and regressed after 12 h.

It should be noted that despite a long period without conducting basic IVIG therapy, none of the patients with CVID did not require its administration before the control points due to the absence of any symptoms of acute respiratory infections.

### 3.2. Assessment of Vaccine Immunogenicity

Given the limited number of patients with CVID (6), first we provide an assessment of the immunogenicity of aQIV in the group of healthy people, and then individually for each CVID patient.

Table 1 and Figure 1 show seroprotection level (HI titer ≥1:40, >70% of participants) in the group of healthy participants vaccinated with aQIV taking into account the initial level of specific antibodies to influenza virus strains.

Before vaccination in the group of initially seropositive participants seroprotection level was 100% to all virus strains. In the post-vaccination period there were no considerable changes, remaining at the level of 95–100%.

In the group of initially seronegative participants, the seroprotection level before vaccination was 0% for all strains. Three weeks after vaccination a statistically significant increase in seroprotection level was up to 70% for strain A/H1N1, 100% for strain A/H3N2, 54% for strain B/Colorado and 75% for strain B/Phuket.

Seroconversion level (4-fold increase in HI titer ≥40% of participants) in vaccinated with aQIV healthy participants, taking into account the initial level of antibodies, is presented in Table 2 and Figure 2.

In the group of initially seronegative patients, the seroconversion level meets the effectiveness criterion to all strains: 70% to the A/H1N1, 100% to the A/H3N2, 46% to B/Colorado, and 63% to B/Phuket.

In the group of initially seropositive patients, the seroconversion level 3 weeks after vaccination was 40% for strain A/H1N1, 40% for strain A/H3N2, 43% for strain B/Colorado, and 39% for strain B/Phuket.

The Table 3 and Figure 3 show geometric mean rate—GMR (≥2.5) in the group of healthy given the initial level of antibodies.

In the group of initially seronegative healthy participants, 3 weeks after vaccination, the GMR for strains A/H1N1, A/H3N2, B/Colorado, B/Phuket was 9.8, 34.3, 3.8 and 5.9, respectively, and for initially seropositive at the same control point of the study, it was registered at the level of 4.8, 2.8, 2.5 and 2.8, respectively.

Geometric mean antibody titers (GMT) in the group of healthy volunteers given the initial level of antibodies are presented in the Table 4 and Figure 4.

Three weeks after vaccination in initially seronegative healthy volunteers the GMT to influenza virus strains significantly increased up to 65.0, 226.3, 27.4 and 51.9 for strains A/H1N1, A/H3N2, B/Colorado, B/Phuket, respectively, and in initially seropositive individuals to the same influenza virus subtypes—up to 452.5, 288.4, 214.9 and 198.0, respectively.

In the post-vaccination period (3 weeks after vaccination) in initially seronegative healthy volunteers the GMT of antibodies to influenza virus strains significantly increased up to 65.0, 226.3, 27.4 and 51.9 for strains A/H1N1, A/H3N2, B/Colorado, B/Phuket, respectively, and in initially seropositive individuals to the same influenza virus subtypes—up to 452.5, 288.4, 214.9 and 198.0, respectively.

An individual analysis of antibodies level to influenza vaccine virus strains in patients with CVID showed that 2 out of 6 patients were seropositive even before vaccination: 1:40 and 1:80 antibody titers to both strains A/H1N1 and A/H3N2 (Table 5). The other 4/6 patients were seronegative for all 4 virus strains included in the influenza vaccine. In the post-vaccination period, after 3 weeks 1/6 patients that were initially seronegative became seropositive (1:80) to strain A/H3N2.

Two out of six CVID patients were seropositive before vaccination: 1:40 and 1:80 antibody titers to both strains A/H1N1 and A/H3N2. The other 4/6 patients were seronegative for all four virus strains included in the influenza vaccine. In the post-vaccination period, after 3 weeks, 1/6 patients that were initially seronegative became seropositive (1:80) to strain A/H3N2. Three months later, the number of seropositive patients increased by one more person, the titer of antibodies was 1:40 to strains A/H1N1, A/H3N2. A tendency to antibodies titers increase 3 months after vaccination was revealed: to strain A/H3N2 from 1:5 to 1:20 in 1/6, to strain B/Colorado in 3/6 patients from 1:5 to 1:20, from 1:5 to 1:10 and from 1:10 to 1:20; for strain B/Phuket there was only an increase in antibodies’ titers in one case, from 1:5 to 1:10.

Three months later the number of seropositive patients increased by one more person, the titer of antibodies was 1:40 to strains A/H1N1, A/H3N2. A tendency to antibodies titers increase 3 months after vaccination was revealed: to strain A/H3N2 from 1:5 to 1:20 in 1/6, to strain B/Colorado in 3/6 patients from 1:5 to 1:20, from 1:5 to 1:10 and from 1:10 to 1:20; for strain B/Phuket only in one case was an increase in antibodies’ titers was from 1:5 to 1:10.

## 4. Discussion

Influenza prophylaxis with the use of quadrivalent vaccines is increasingly being introduced into healthcare practice. Vaccine adjuvants have advantages in activation not only humoral, but also cellular mechanisms of the immune response with improved clinical and immunological effects of vaccination. Despite the existence of numerous adjuvants, there is a limited number of candidates, involved in the development of vaccines, due to the safety of their usage.

A family of influenza vaccines with azoximer bromide as an adjuvant to enhance the humoral immune response (Grippol, Grippol plus, MonoGrippol) has been used in Russia for more than 20 years within the framework of the National Schedule of Preventive Vaccinations. Azoximer bromide induces dendritic cells maturation with increased expression of co-stimulatory molecules CD80/86 and ICOSLG, that are necessary for subsequent activation of follicular T-cells, which are in a key link with production of specific high-affinity antibodies by B-cells [25,26]. Long term use of the trivalent Grippol vaccines in healthcare practice proved their safety and immunogenicity, despite the decrease in the amount of antigens in total to 15 μg versus 45 μg in a single standard non-adjuvant vaccination dose [27,28]. Registration of the vaccine Grippol^®^ Quadrivalent (aQIV) necessitates additional studies to evaluate its effectiveness not only in healthy people, but also in patients with various diseases, among which the most complex are patients with PID. In our study, we evaluated the safety and immunogenicity of aQIV in healthy adults and estimated the effectiveness of vaccination in patients with common variable immune deficiency.

During 2018–2019 flu season it was possible to vaccinate just six patients with CVID. One of the main reasons is that CVID, a primary immunodeficiency, belongs to the group of orphan diseases (the term “orphan disease” is used to designate diseases that affect only small numbers of individuals; as a condition that affects fewer than 200,000 people nationwide.). That is why almost all investigations in the database Pubmed concerning vaccination, especially influenza vaccination, and the study of post-vaccination immune response include very small amount of participants (2–8 patients) [29,30,31,32,33]. Just one study, which was held in 2011 and published only in 2018, involved 48 CVID patients who were vaccinated during the H1N1 influenza pandemic with monovalent adjuvant influenza vaccine, which allowed the involvement of a large amount of patients.

The first characteristic studied in post-vaccination period was an assessment of clinical tolerance of aQIV in healthy people and patients with CVID, and only one patient developed a local reaction in the form of hyperemia and edema of soft tissues within 3 days, which disappeared without any medication. It is important to note that vaccination did not affect patients with CVID on their daily well-being and through the course of the disease.

Evaluation of the aQIV immunogenicity in healthy people, given the pre-vaccination level of antibodies, with the distribution of participants into seronegative and seropositive groups before vaccination allows us to argue the ability of the immune system to respond by the production of specific antibodies to influenza virus strains in case of initial or repeated administration of the vaccine.

A study of the antibodies titer 3 weeks after immunization in 32 healthy participants showed that for strains A/H1N1, A/H3N2 and B/Phuket seroprotection level among initially seronegative participants meets the criterion of CHMP effectiveness (at least 70%). Despite a significant increase in the seroprotection level among initially seronegative participants, the proportion of seropositive to strain A/H1N1 and strain B/Colorado remains less than the same indicator in the group of initially seropositive patients.

Administration of aQIV in healthy adults leads to the formation of specific antibodies, the seroconversion level meets the CHMP criterion of effectiveness (at least 40%) with the exception of the B/Phuket strain—where seroconversion level is on the verge of a threshold value. The level of seroconversion to strain A/H3N2 is significantly higher in the group of initially seronegative patients (in 2.5 times: 100% versus 40%); for other strains, there were no statistically significant differences in seroconversion level.

Analysis of the results of vaccinated healthy adults 3 weeks after immunization showed that GMR meets the CHMP criteria for the effectiveness for all strains regardless of the initial level of antibodies. However, it should be noted that GMR is higher in the group of initially seronegative compared with seropositive in relation to strain A/H3N2 (34.3 versus 2.8) and B/Phuket (5.9 versus 2.8).

There is a statistically significant increase in GMT to all virus strains in post-vaccination period in healthy participants, regardless of the initial antibody titer. The largest increase is observed among the initially seronegative participants for strain A/H1N1—34.2 times. As a result, GMT to this strain one month after vaccination did not differ statistically significant among initially seronegative and seropositive. Despite the significant increase in the post-vaccination period of specific antibodies to the all strains, as well as to A/H1N1 strain (9.2 vs. 4.8), B/Colorado (3.7 vs. 2.4) in initially seronegative healthy volunteers, the level of GMT to A/H1N1, B/Colorado and B/Phuket strains is higher among initially seropositive both before vaccination and 3 weeks after.

Initially, before vaccination, regardless of the virus strain, the proportion of seropositive participants between the groups of healthy adults and patients with CVID did not differ to a statistically significant extent, probably the differences were not revealed due to a small cohort of patients. It should be noted that both before and 3 weeks after vaccination, the proportion of seropositive patients (2 out of 6) for strains A/H1N1, A/H3N2 is the same, without a subsequent increase in specific antibodies. However, in one patient, an increase in antibodies to protective values was detected for strain A/H3N2 from 1:20 to 1:80.

Regarding B lineage strains, all CVID patients (6) were seronegative before vaccination, but 3 weeks after the administration of aQIV there was some tendency to increase in antibodies to B/Colorado in 1/6, and 3 months after in 3/6, patients (≥2-fold increase in the titer of antibodies). Moreover, in one patient, the antibody growth was by 4 times. Only 1/6 patient showed an increase in HI antibodies titer to B/Phuket strain from 1:5 to 1:10 3 months after immunization, while it is important to notice that for patients with CVID, a conditional indicator of the effectiveness of vaccination is an 2–4-fold increase in post-vaccination antibody levels. Interestingly, 3 months after immunization, there were 3/6 patients with a protective level (1:40) of antibodies to the A/H1N1 strain, and 5/6 patients to A/H3N2 (one of them showed an increase in antibodies titer from 1:5 to 1:20 (4-fold increase—seroconversion level)).

## 5. Conclusions

This study of aQIV in healthy adults, in which the immunotropic drug (azoximer bromide) was used as an adjuvant, demonstrated that the vaccine promotes the formation of specific immunity to both strains of influenza A and B types, regardless of the presence or absence of antibodies to the vaccine strains. Therefore, the vaccine can be used both for the primary and re-vaccination of people previously vaccinated against influenza. At the same time, it was shown that in the post-vaccination period, in naive individuals, the increase in specific antibodies is higher than the level of antibodies in initially seropositive people. In addition, re-vaccination with aQIV in adults vaccinated against influenza in previous seasons did not lead to a decrease in the level of post-vaccination antibodies. Thus, aTIV4 with a 3-fold reduction in the amount of antigens in one vaccination dose effectively induces an immune response comparable with non-adjuvant quadrivalent influenza vaccines containing 60 μg of antigens. Moreover, the use of aQIV may provide broader epidemiological protection against influenza. The absence of the development of unusual adverse reactions in the post-vaccination period is encouraging for its prospective use in children, including those with allergic pathology, among whom the frequency of local reactions may be registered more often than in the general pediatric population [28].

Of great importance are the data obtained during the vaccination of patients with CVID, showing that the standard scheme of administering aQIV is not able to form specific antibodies on the early stages, although it remains unclear how cellular mechanisms are activated in the post-vaccination period in patients with defects in humoral immunity. Recent studies indicate that administration of adjuvant trivalent subunit influenza vaccine increases the amount of activated dendritic cells that persist even a month after immunization, while subunit and split influenza vaccines activate these cells only within 7 days after vaccination. Moreover, the multiplicity of increase in the number of dendritic cells after adjuvant vaccine administration is almost twice higher than in the case of the non-adjuvant vaccine [34]. Adjuvant increases the speed of the migration of dendritic cells to the lymph nodes. It also accelerates the maturation of dendritic cells. At the same time, pronounced activation of both cellular and humoral immunity is observed [35,36,37]. Moreover, adjuvant influenza vaccine activates additional receptors that are responsible for bacterial sensitization and the recognition of bacterial antigens. In addition, non-specific antiviral immunity is activated [38,39].

Studies have shown that in people vaccinated with the adjuvant influenza vaccine, the respiratory infection rate is about 9%; in those vaccinated with a subunit vaccine, it is 14%; and in those with a split vaccine, it is 11% [40]. Therefore, the absence of specific antibodies to influenza virus strains in patients with CVID in protective values does not mean that they did not form protection against influenza with the involvement of T-cell mechanisms of the immune response. This is also indicated by the results of experiments held by other authors [32,39,41,42,43]. Moreover, the synthesis of specific antibodies to the influenza virus strains is possibly delayed in patients with CVID. This may be proven by the low levels of antibodies measured 3 weeks after aQIV administration.

Thus, the expansion of vaccination programs with the use of aQIV containing immunoadjuvant is surely promising among the healthy adult population. However, in patients with PID, the search for new vaccination regimens based on scientific evidence should be continued.

## Figures and Tables

**Figure 1 vaccines-08-00640-f001:**
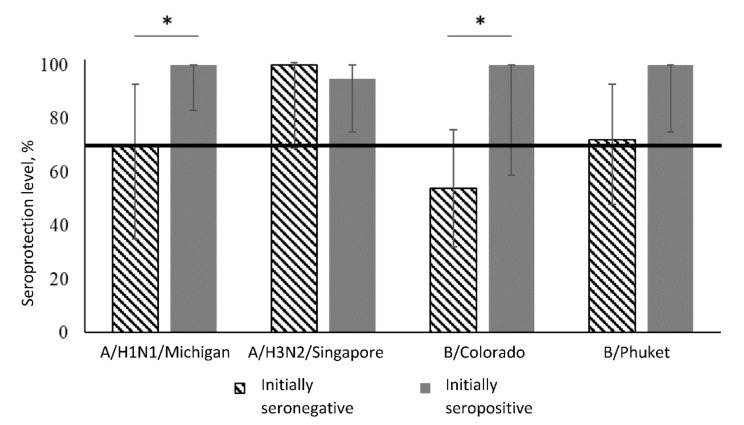
Seroprotection level 1 month after immunization given the initial level of antibodies in the group of healthy participants vaccinated with aQIV. Notes: *—statistically significant differences between groups

**Figure 2 vaccines-08-00640-f002:**
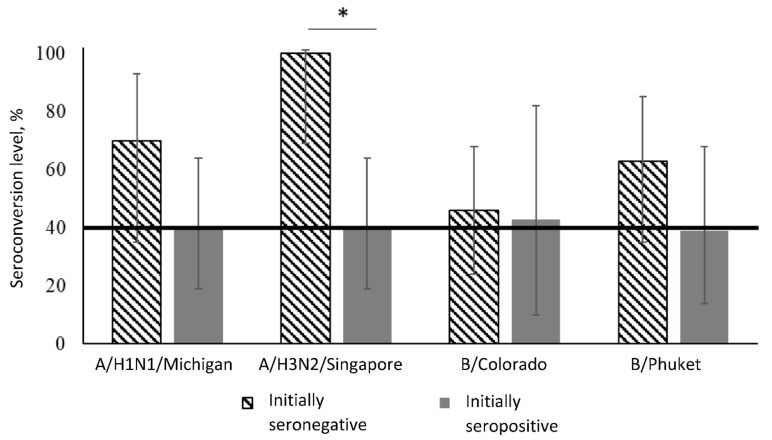
Seroconversion level 3 weeks after immunization given the initial level of antibodies in the group of healthy participants. Notes: *—statistically significant differences between groups.

**Figure 3 vaccines-08-00640-f003:**
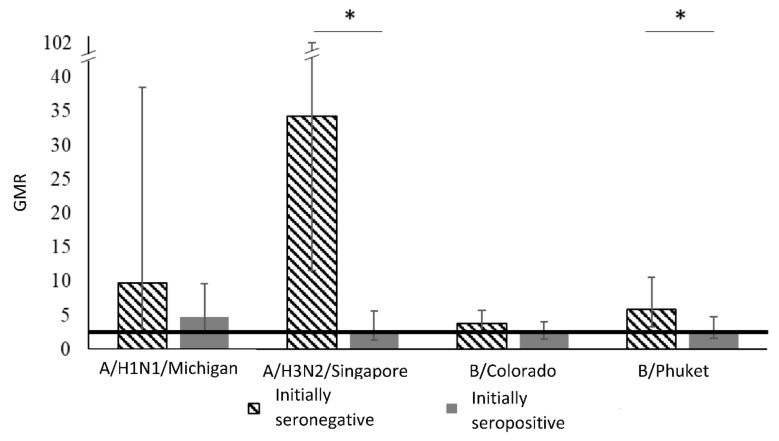
GMR given the initial level of antibodies in the group of healthy participants. Notes: *—statistically significant differences between groups.

**Figure 4 vaccines-08-00640-f004:**
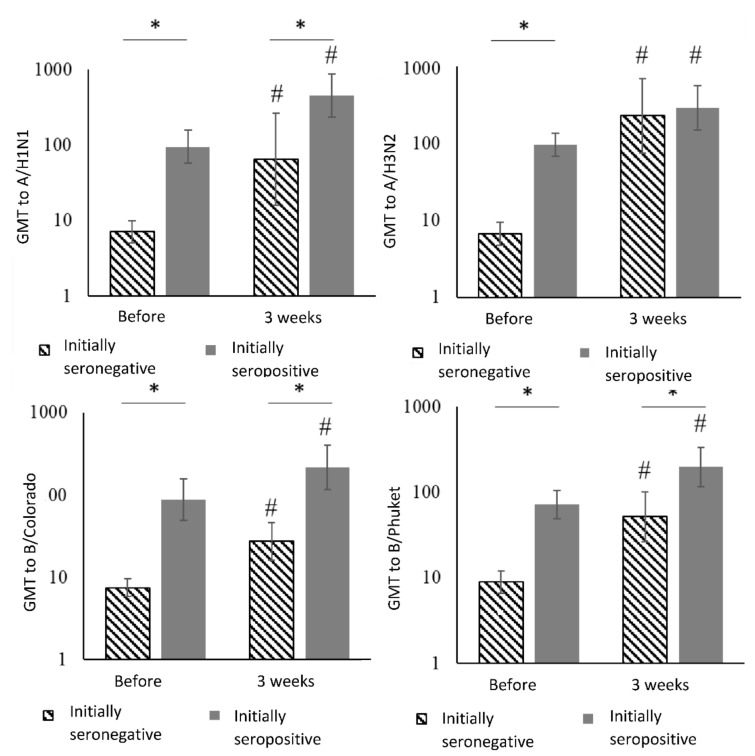
Geometric mean antibody titers (GMT) in the group of healthy participants given the initial level of antibodies. Notes: *—statistically significant differences between groups, #—statistically significant differences compared with the initial level.

**Table 1 vaccines-08-00640-t001:** Seroprotection levels given the initial level of antibodies in the group of healthy participants.

Virus Strain	Period	Initially Seronegative			Initially Seropositive			Between Groups ^1^
		People	%	95%CI	People	%	95%CI	
A/H1N1	Before vaccination	0/12	0	(0–26)	20/20	100	(83–100)	***p* < 0.001**
	After 3 weeks	7/10	70	(35–93)	20/20	100	(83–100)	***p* = 0.03**
Dynamics analysis ^2^		***p* = 0.02**			*p* = 1.00			–
A/H3N2	Before vaccination	0/10	0	(0–31)	22/22	100	(85–100)	***p* < 0.001**
	After 3 weeks	10/10	100	(69–100)	19/20	95	(75–100)	*p* = 1.00
Dynamics analysis		***p* = 0.002**			*p* = 1.00			–
B/Colorado	Before vaccination	0/24	0	(0–14)	7/7	100	(59–100)	***p* < 0.001**
	After 3 weeks	12/22	54	(32–76)	7/7	100	(59–100)	***p* = 0.03**
Dynamics analysis		***p* < 0.001**			*p* = 1.00			–
B/Phuket	Before vaccination	0/18	0	(0–19)	13/13	100	(75–100)	***p* < 0.001**
	After 3 weeks	12/16	72	(48–93)	13/13	100	(75–100)	*p* = 0.08
Dynamics analysis		***p* < 0.001**			*p* = 1.00			–

Notes: ^1^—Chi-Square test was used, in case of cells in the table with expected frequencies of less than 5%, Fisher’s exact test was used, ^2^—McNemar’s criterion was used.

**Table 2 vaccines-08-00640-t002:** Seroconversion level given the initial level of antibodies in the group of healthy participants.

Groups		Virus Strains			
		A/H1N1	A/H3N2	B/Colorado	B/Phuket
Initially seronegative	People	7/10	10/10	10/22	10/16
	%	70	100	46	63
	95% CI	(35–93)	(69–100)	(24–68)	(35–85)
Initially seropositive	People	8/20	8/20	3/7	5/13
	%	40	40	43	39
	95% CI	(19–64)	(19–64)	(10–82)	(14–68)
Comparison between groups ^1^		*p* = 0.12	*p* = 0.002	*p* = 1.00	*p* = 0.20

Notes: ^1^—the Chi-Square test was applied, in the case of cells in the table with expected frequencies of less than 5%, the exact Fisher test was used.

**Table 3 vaccines-08-00640-t003:** Geometric mean rate (GMR) in the group of healthy given the initial level of antibodies in the group of healthy participants.

Groups		Virus Strains			
		A/H1N1	A/H3N2	B/Colorado	B/Phuket
Initially seronegative	GMR	9.8	34.3	3.8	5.9
	95%CI	(2.5–38.5)	(11.6–101.3)	(2.4–5.8)	(3,3–10.6)
Initially seropositive	GMR	4,8	2,8	2,5	2,8
	95%CI	(2.4–9.6)	(1.4–5.6)	(1,5–4.0)	(1.6–4.8)
Comparison between groups ^1^		0.30	<0.001	0.31	0.05

Notes: ^1^—Mann–Whitney test was used.

**Table 4 vaccines-08-00640-t004:** GMT in the group of healthy participants given the initial level of antibodies.

Virus strains	Period	Initially Seronegative		Initially Seropositive		Between Groups ^1^
		GMT	95%CI	GMT	95%CI	
A/H1N1	Before vaccination	7.1	(5.0–10.0)	95.1	(57.5–157.4)	*p* < 0.001
	After 3 weeks	65.0	(16.0–264.4)	452.5	(235.5–869.5)	*p* = 0.004
	Dynamics analysis ^2,3^	*p* = 0.004		*p* < 0.001		–
A/H3N2	Before vaccination	6.6	(4.7–9.3)	96.6	(68.5–136.4)	*p* < 0.001
	After 3 weeks	226.3	(75.1–681.5)	288.4	(147.9–562.4)	*p* = 0.67
	Dynamics analysis	*p* < 0.001		*p* = 0.005		–
B/Colorado	Before vaccination	7.5	(5.9–9.6)	88.3	(49.6–157.3)	*p* < 0.001
	After 3 weeks	27.4	(16.3–46.1)	214.9	(115.6–403.4)	*p* < 0.001
	Dynamics analysis	*p* < 0.001		*p* = 0.004		–
B/Phuket	Before vaccination	8.9	(6.6–12.0)	71.9	(49.4–104.8)	*p* < 0.001
	After 3 weeks	51.9	(26.5–101.6)	198.0	(117.3–334.4)	*p* = 0.003
	Dynamics analysis	*p* < 0.001		*p* = 0.002		–

Notes: ^1^—Student criterion was used, ^2^—Student criterion was used for paired samples, ^3^—all criteria were counted on pre-logarithmized data.

**Table 5 vaccines-08-00640-t005:** Seroprotection level in the group of patients with common variable immunodeficiency (CVID) vaccinated with adjuvant QIV (aQIV).

Virus Strains	Period	Patient 1	Patient 2	Patient 3	Patient 4	Patient 5	Patient 6
A/H1N1	Before vaccination	40	20	20	20	20	80
	After 3 weeks After 3 months	40 40	20 40	20 20	20 20	10 10	40 40
A/H3N2	Before vaccination	40	20	5	10	10	80
	After 3 weeks After 3 months	40 40	20 40	5 20	80 40	5 5	80 40
B/Colorado	Before vaccination	10	5	5	5	5	20
	After 3 weeks After 3 months	10 20	5 10	5 5	10 20	5 5	10 10
B/Phuket	Before vaccination	20	10	10	5	10	20
	After 3 weeks After 3 months	10 10	10 10	10 10	5 10	5 5	20 20

## Data Availability

The authors declare that the data supporting the findings of this study are available within the paper and its supplementary information files.

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
