# Peer review of "Immunogenicity and Safety of the Quadrivalent Adjuvant Subunit Influenza Vaccine in Seropositive and Seronegative Healthy People and Patients with Common Variable Immunodeficiency"

_vaccines, 2020, doi:10.3390/vaccines8040640_

Round 1

Reviewer 1 Report

The authors evaluate antibody responses to the clinically approved Grippol Quadrivalent influenza vaccine. They assess for any adverse reactions over the short follow-up period (3 weeks) of their study and for standard vaccine antibody responses in 32 healthy patients and 6 CVID patients (B cell immunodeficiency). I have a number of concerns in regard to this paper.

Major:

1) The paper is presented as a Stage IV trial. The timeframe that the patients are covered and the small size of the cohort do not lend any valuable post-marketing information regarding the healthy population that would not have already been gleaned in earlier trials. This data would be better combined with other Stage IV trials and extended for greater periods of follow-up.

2) While the authors state ethics approval for Stage IV trial it is unclear if this ethics approval included the removal of CVID patients from IVIG therapy and the consequent risks to their well-being.

3) The CVID data is the most novel and interesting information in this paper. However, it is extremely weak. More effort should have been made to characterize the genetic basis of immunodeficiency in the patients, to expand enrolment in study to beyond 6 patients and to investigate the mechanism of the mild and rare delayed responses in some of the patients. The authors postulate that adujvant may result in better T cell responses and potential protection but do no assays to evaluate this hypothesis.

Minor:

1) While I understand English may not be native to the authors (and the English is actually fairly good) the paper is written rather poorly. Words are missing and some sections, particularly the results, are choppy and difficult to read. Figure legend are sparse and require the reader to flip around in the paper a lot to know what they are looking at. 

Author Response

Thanks a lot for your review!

Major:

  • Yes, you are right that immunogenicity in healthy cohort doesn’t lend valuable post-marketing information but in our study we decided that it was necessary as a control group. Besides, for the first time immunogenicity evaluation of the influenza vaccine was made according to the last revision of the Guideline on clinical evaluation of vaccines from 26 April 2018 of the Committee on Human Medicinal Products (CHMP): new criterion for adult patients such as “post-vaccination seroprotection and seroconversion levels separately for those who were seronegative or seropositive at study baseline” was used.

  • CVID patients were provided with study diaries, as well as constant communication through a daily telephone survey recording the frequency of local and systemic reactions within 7 days after vaccination and the next 14 days recording well-being, of course. During that period of time 4/6 had some problems with the provision of monthly immunoglobulins (2-3 weeks of drug delay) in Moscow. So in this situation, it can even be said that CVID patients have benefited during the period of insecurity without IVIG therapy. Moreover, before the study, it was discussed that in the event of an infectious disease within 3 weeks of supervision after vaccination, we were ready to administer immediately a basic individual dose of immunoglobulins for each of these 6 participants. Fortunately, despite a long period without conducting basic IVIG therapy, none of the patients with CVID did not required its administration before the control points due to the absence of any symptoms of acute respiratory infections.

  • Thank you very much for assessment of our study! You are right about the number of CVID patients, during the 2019-2020 flu season we have vaccinated 15 CVID patients using another schemes of vaccination and evaluation of T-cell immunity, unfortunately, the results are not ready yet.

But during 2018-2019 flu season it was possible to vaccinate just 6 patients with CVID because of the profile of our center, we hospitalize patients from the whole Russian Federation territory with exacerbation of chronic bronchopulmonary diseases as usual, that are often needed even in additional replacement IVIG therapy besides their basic regular monthly IVIG infusions. So that is why we could not vaccinate them (inclusion criteria - "IVIG therapy no later than 28 days before vaccination and no earlier than 21 days after it, that is, a break between two subsequent administrations of immunoglobulins for at least 7 weeks") during the period of hospital treatment, that could provide significantly more participants among CVID patients.

The next reason is that CVID, primary immunodeficiency, belongs to the group of orphan diseases (the term "orphan disease" is used to designate diseases that affect only small numbers of individuals; as a condition that affects fewer than 200,000 people nationwide (www.fda.gov)). That is why in fact almost all investigations in the database Pubmed concerning vaccination, especially influenza vaccination, and the study of post-vaccination immune response include very small amount of participants (2-8):

  1. van Assen S, Holvast A, Telgt DS, Benne CA, de Haan A, Westra J, et al. Patients with humoral primary immunodeficiency do not develop protective anti-influenza antibody titers after vaccination with trivalent subunit influenza vaccine. Clin Immunol (Orlando, Fla). 2010; 136(2):228–35. https://doi.org/10.1016/j.clim.2010.03.430.

  1. Hanitsch LG, Lobel M, Mieves JF, Bauer S, Babel N, Schweiger B, et al. Cellular and humoral influenza-specific immune response upon vaccination in patients with common variable immunodeficiency and unclassified antibody deficiency. Vaccine. 2016; 34(21): 2417–23. https://doi.org/10.1016/j.vaccine.2016.03.091.

  1. van Assen S, de Haan A, Holvast A, Horst G, Gorter L, Westra J, et al. Cell-mediated immune responses to inactivated trivalent influenza-vaccination are decreased in patients with common variable immunodeficiency. Clin Immunol (Orlando, Fla). 2011; 141(2):161–8. https://doi.org/10.1016/j.clim.2011.07.004.

  1. Pedersen G, Halstensen A, Sjursen H, Naess A, Kristoffersen EK, Cox RJ. Pandemic influenza vaccination elicits influenza-specific CD4+ Th1-cell responses in hypogammaglobulinaemic patients: four case reports. Scand J Immunol. 2011; 74(2): 210–8. https:// doi.org/10.1111/j.1365-3083.2011.02561.x.

  1. Manuel O, Pascual M, Hoschler K, et al. Humoral response to the influenza A H1N1/09 monovalent AS03-adjuvanted vaccine in immunocompromised patients. Clin Infect Dis 2011; 52:248-56. doi: 10.1093/cid/ciq104.

Just one study which was held in 2011 and published only in 2018 involved 48 CVID patients who were vaccinated during the H1N1 influenza pandemic with monovalent adjuvant influenza vaccine, that allowed to involve large amount of patients.

If it’ll be interesting and informative, we can provide and add the material about

accompanying illnesses (interstitial lung disease, chronic sinusitis, hepatomegaly, slenomegaly and others) in participated 6 CVID patients!

Minor:

  • Table legend: 615-618 - Before vaccination in the group of initially seropositive participants seroprotection level was 100% to all virus strains. In the post-vaccination period there were no considerable changes, remaining at the level of 95-100%. In the group of initially seronegative seroprotection level before vaccination was 0% for all strains. 3 weeks after vaccination a statistically significant increase in seroprotection level was up to 70% for strain A/H1N1, 100% for strain A/H3N2, 54% for strain B/Colorado and 75% for strain B/Phuket.
  • Figure legend: 633-635 - Despite a significant increase in the seroprotection level among initially seronegative participants, the proportion of seropositive to strain A/H1N1 and strain B/Colorado remains less than the same indicator in the group of initially seropositive.
  • Table legend: 643-646 - In the group of initially seronegative patients seroconversion level meets the effectiveness criterion to all strains: 70% to the A/H1N1, 100% to the A/H3N2, 46% to B/Colorado and 63% to B/Phuket. In the group of initially seropositive patients seroconversion level 3 weeks after vaccination was 40% for strain A/H1N1, 40% for strain A/H3N2, 43% for strain B/Colorado and 39% for strain B/Phuket.
  • Figure legend: 654-657 - Seroconversion level meets the CHMP criterion of effectiveness (at least 40%) with the exception of B/Phuket strain – seroconversion level is on the verge of a threshold value. The level of seroconversion to strain A/H3N2 is significantly higher in the group of initially seronegative (in 2.5 times: 100% versus 40%), for other strains there were no statistically significant differences in seroconversion level.
  • Table legend: 663-665 - In the group of initially seronegative healthy participants 3 weeks after vaccination GMR for strains A/H1N1, A/H3N2, B/Colorado, B/Phuket was 9,8, 34,3, 3,8 and 5,9, respectively, for initially seropositive it was at the level of 4,8, 2,8, 2,5 and 2,8, respectively.
  • Figure legend: 672-674 - GMR meets the CHMP criteria for the effectiveness for all strains regardless of the initial level of antibodies. But it should be noted that GMR is higher in the group of initially seronegative compared with seropositive in relation to strain A/H3N2 (34.3 versus 2.8) and B/Phuket (5.9 versus 2.8).
  • Table legend: 682-685 - 3 weeks after vaccination in initially seronegative healthy volunteers the GMT to influenza virus strains significantly increased up to 65,0, 226,3, 27,4 and 51,9 for strains A/H1N1, A/H3N2, B/Colorado, B/Phuket, respectively, and in initially seropositive individuals to the same influenza virus subtypes - up to 452,5, 288,4, 214,9 and 198,0, respectively.
  • Figure legend: 712-718 - There is a statistically significant increase in GMT to all virus strains in post-vaccination period in healthy participants, regardless of the initial antibody titer. The largest increase is observed among the initially seronegative participants for strain A/H1N1 - in 34.2 times. GMT to this strain one month after vaccination did not differ statistically significant among initially seronegative and seropositive. Despite the significant increase in post-vaccination period of specific antibodies to all strains, the level of GMT to A/H1N1, B/Colorado and B/Phuket strains is higher among initially seropositive both before vaccination and 3 weeks after.
  • Table legend: 727-734 - 2/6 CVID patients were seropositive before vaccination: 1:40 and 1:80 antibody titers to both strains A/H1N1 and A/H3N2. The other 4/6 patients were seronegative for all 4 virus strains included in the influenza vaccine. In the post-vaccination period after 3 weeks 1/6 patient initially seronegative became seropositive (1:80) to strain A/H3N2. Three months later the number of seropositive patients increased by one more person, the titer of antibodies was 1:40 to strains A/H1N1, A/H3N2. A tendency to antibodies titers increase 3 months after vaccination was revealed: to strain A/H3N2 from 1:5 to 1:20 in 1/6, to strain B/Colorado in 3/6 patients from 1:5 to 1:20, from 1:5 to 1:10 and from 1:10 to 1:20; for strain B/Phuket only in one case was an increase in antibodies’ titers was from 1:5 to 1:10.

Reviewer 2 Report

The study proposes use of an adjuvant Polyoxidonium or azoximer bromide to test Quadrivalent (including both B strains of Influenza that are commonly used as either one for regular vaccine) Influenza vaccine immunogenicity and safety. Authors clearly state the importance of the study, which the reviewer agree.

However, the title of the MS also points out testing the platform in CVID. Despite their attempt to compare Trivalent vs quadrivalent, application of such is quite discouraged for CVID with limited samples. The reviewer is wondering if they can formulate the title in a different way or include more patient samples with CVID. This is the major issue with the MS.

Author Response

Thank you very much for assessment of our study!

You are right about the number of CVID patients, during the 2019-2020 flu season we have vaccinated 15 CVID patients using another schemes of vaccination, unfortunately, the results are not ready yet.

But during 2018-2019 flu season it was possible to vaccinate just 6 patients with CVID because of the profile of our center, we hospitalize patients from the whole Russian Federation territory with exacerbation of chronic bronchopulmonary diseases as usual, that are often needed even in additional replacement IVIG therapy besides their basic regular monthly IVIG infusions. So that is why we could not vaccinate them (inclusion criteria - "IVIG therapy no later than 28 days before vaccination and no earlier than 21 days after it, that is, a break between two subsequent administrations of immunoglobulins for at least 7 weeks") during the period of hospital treatment, that could provide significantly more participants among CVID patients.

The next reason is that CVID, primary immunodeficiency, belongs to the group of orphan diseases (the term "orphan disease" is used to designate diseases that affect only small numbers of individuals; as a condition that affects fewer than 200,000 people nationwide (www.fda.gov)). That is why in fact almost all investigations in the database Pubmed concerning vaccination, especially influenza vaccination, and the study of post-vaccination immune response include very small amount of participants (2-8):

  1. van Assen S, Holvast A, Telgt DS, Benne CA, de Haan A, Westra J, et al. Patients with humoral primary immunodeficiency do not develop protective anti-influenza antibody titers after vaccination with trivalent subunit influenza vaccine. Clin Immunol (Orlando, Fla). 2010; 136(2):228–35. https://doi.org/10.1016/j.clim.2010.03.430.

  1. Hanitsch LG, Lobel M, Mieves JF, Bauer S, Babel N, Schweiger B, et al. Cellular and humoral influenza-specific immune response upon vaccination in patients with common variable immunodeficiency and unclassified antibody deficiency. Vaccine. 2016; 34(21): 2417–23. https://doi.org/10.1016/j.vaccine.2016.03.091.

  1. van Assen S, de Haan A, Holvast A, Horst G, Gorter L, Westra J, et al. Cell-mediated immune responses to inactivated trivalent influenza-vaccination are decreased in patients with common variable immunodeficiency. Clin Immunol (Orlando, Fla). 2011; 141(2):161–8. https://doi.org/10.1016/j.clim.2011.07.004.

  1. Pedersen G, Halstensen A, Sjursen H, Naess A, Kristoffersen EK, Cox RJ. Pandemic influenza vaccination elicits influenza-specific CD4+ Th1-cell responses in hypogammaglobulinaemic patients: four case reports. Scand J Immunol. 2011; 74(2): 210–8. https:// doi.org/10.1111/j.1365-3083.2011.02561.x.

  1. Manuel O, Pascual M, Hoschler K, et al. Humoral response to the influenza A H1N1/09 monovalent AS03-adjuvanted vaccine in immunocompromised patients. Clin Infect Dis 2011; 52:248-56. doi: 10.1093/cid/ciq104.

Just one study which was held in 2011 and published only in 2018 involved 48 CVID patients who were vaccinated during the H1N1 influenza pandemic with monovalent adjuvant influenza vaccine, that allowed to involve large amount of patients.

Round 2

Reviewer 1 Report

Thank you for providing additional information on your figures and tables. Although not standard in form it is more helpful to the reader. 

It would be useful to include the information you provided regarding the ethical measures taken to ensure the safety of the CVID patients in the methods (if you cannot just outright confirm you have some sort of formal Ethics Board approval for their specific inclusion into the study). Additionally, the limitations of the CVID portion of the study should be discussed more thoroughly in the results and discussion sections. 

Author Response

Thanks a lot for your review one more time!

Minor:

Lines 236-246: They were provided with study diaries, as well as constant communication through a daily telephone survey recording the frequency of local and systemic reactions within 7 days after vaccination, concerning CVID patients – 14 days more, recording well-being. During that period of time 4/6 had some problems with the provision of monthly immunoglobulins (2-3 weeks of drug delay). So it can even be said that CVID patients have benefited during the period of insecurity without IVIG therapy. Moreover, before the study, it was discussed that in the event of an infectious disease within 3 weeks of supervision after vaccination, we were ready to administer immediately a basic individual dose of immunoglobulins for each of these 6 participants. Fortunately, despite a long period without conducting basic IVIG therapy, none of the patients with CVID did not required its administration before the control points due to the absence of any symptoms of acute respiratory infections.

Lines 356-364: During 2018-2019 flu season it was possible to vaccinate just 6 patients with CVID. One of the main reasons is that CVID, primary immunodeficiency, belongs to the group of orphan diseases (the term "orphan disease" is used to designate diseases that affect only small numbers of individuals; as a condition that affects fewer than 200,000 people nationwide.). That is why in fact almost all investigations in the database Pubmed concerning vaccination, especially influenza vaccination, and the study of post-vaccination immune response include very small amount of participants (2-8 patients) [29-33]. Just one study which was held in 2011 and published only in 2018 involved 48 CVID patients who were vaccinated during the H1N1 influenza pandemic with monovalent adjuvant influenza vaccine, that allowed to involve large amount of patients.

Reviewer 2 Report

I would strongly suggest authors to put more on CVID results on sample size limitation in the discussion sections (The answers given to the reviewers as response).

Author Response

Thanks a lot for your review one more time!

Lines 236-246: They were provided with study diaries, as well as constant communication through a daily telephone survey recording the frequency of local and systemic reactions within 7 days after vaccination, concerning CVID patients – 14 days more, recording well-being. During that period of time 4/6 had some problems with the provision of monthly immunoglobulins (2-3 weeks of drug delay). So it can even be said that CVID patients have benefited during the period of insecurity without IVIG therapy. Moreover, before the study, it was discussed that in the event of an infectious disease within 3 weeks of supervision after vaccination, we were ready to administer immediately a basic individual dose of immunoglobulins for each of these 6 participants. Fortunately, despite a long period without conducting basic IVIG therapy, none of the patients with CVID did not required its administration before the control points due to the absence of any symptoms of acute respiratory infections.

Lines 356-364: During 2018-2019 flu season it was possible to vaccinate just 6 patients with CVID. One of the main reasons is that CVID, primary immunodeficiency, belongs to the group of orphan diseases (the term "orphan disease" is used to designate diseases that affect only small numbers of individuals; as a condition that affects fewer than 200,000 people nationwide.). That is why in fact almost all investigations in the database Pubmed concerning vaccination, especially influenza vaccination, and the study of post-vaccination immune response include very small amount of participants (2-8 patients) [29-33]. Just one study which was held in 2011 and published only in 2018 involved 48 CVID patients who were vaccinated during the H1N1 influenza pandemic with monovalent adjuvant influenza vaccine, that allowed to involve large amount of patients.
